# Olivine's high radiative conductivity increases slab temperature by up to 200K

Enrico Marzotto [1,2] ✉, Alexander Koptev[1,3], Sergio Speziale[1], Monika Koch-Müller[1], Nada Abdel-Hak [1,4], Sarah B. Cichy[1,2] & Sergey S. Lobanov[1,2] ✉

The thermal evolution of slabs governs their subduction dynamics and the transport of water into Earth's interior. However, current slab subduction models often neglect the contribution of radiative thermal conductivity (i.e., heat transport by light) due to the limited constraints on the opacity of minerals at high pressure ($P$) and temperature ($T$). Here, using optical experiments at high $P, T$ conditions, we show that the radiative contribution accounts for ~40% of the total heat transport in olivine, the dominant mineral of the upper mantle. Using 2D thermo-kinematic modelling, we quantify the effect of radiative thermal conductivity on slab temperature exploring different ages and subduction velocities. When radiative heating is included, slabs temperatures are ~100−200 $K$ higher than in the models that ignore this contribution. Consequently, water-bearing minerals can reach the Mantle Transition Zone (~410−660 $km$) only in old slabs (>60 $Myrs$) and/or at high subduction velocities (≥10 $cm/year$).

Earth is the only known planet featuring plate tectonics[1], whose main driving force is the gravitational pull of the oceanic lithosphere sinking into the mantle, namely: slab subduction[2,3]. Water stored in the slab is fundamental to the onset of subduction, and is thus crucial to sustain plate tectonics[4], by reducing solidus temperature[5] and mechanical strength[6] of the rocks. Numerical models[7] suggest that ~2/3 of the water stored inside the slab is released before reaching 230 $km$ of depth due to the decomposition of temperature sensitive hydrous minerals[8]. Dehydration processes trigger flux melting[5], leading to arc volcanism[9], and dehydration embrittlement[6], which is responsible for intermediate depth seismicity[10] (70−300 $km$). Some hydrous minerals, however, can survive longer inside the cold core of the slab[11], potentially forming a deep water reservoir[8] in the region extending from ~410 to ~660 $km$ of depth, also known as Mantle Transition Zone (MTZ)[12]. Modelling slab thermal evolution and the depth of water release in the mantle is therefore essential for understanding the deep-Earth water cycle and plate tectonics.

Slab thermal evolution $\partial T/\partial t$ is usually computed with numerical models[13], using Fourier's Law[14] of heat conduction $Q = -\Lambda \nabla T$. This equation describes temperature changes as a function of the amount of thermal energy that enters or leaves an object i.e., the heat flux $Q\ [Wm^{-2}]$. The key parameter in this equation is the thermal conductivity, $\Lambda\ [Wm^{-1}K^{-1}]$, as it quantifies the amount of heat transported. In electrical insulators, such as silicate minerals, the heat propagates as acoustic vibrations of the crystal lattice (phonons)[15], and as optical radiation (photons)[16]. The total thermal conductivity $\Lambda$ is the sum of the lattice ($\Lambda_{lat}$) and radiative ($\Lambda_{rad}$) components: $\Lambda = \Lambda_{lat} + \Lambda_{rad}$. Importantly, the physical properties of subducting rocks change when the slab is subjected to high pressure $P[Pa]$ and temperature $T[K]$ conditions of the Earth's interior[17]. The effects of increasing $P, T$ on lattice heat transport of major upper mantle minerals are well known: $P$ increases $\Lambda_{lat}$ by increasing mean phonon velocity[18], whereas $T$ reduces $\Lambda_{lat}$ by increasing phonons collisions[19]. In contrast, radiative heat transport increases rapidly with $T^3$ (Stephan-Boltzmann's law)[16], but decreases with increasing opacity of minerals, which itself is a function of $P$ and $T^{20-37}$. However, most geodynamic models do not consider the effects of pressure and temperature on lattice and radiative heat transport[13], generally assuming a constant $\Lambda_{lat}$ and neglecting $\Lambda_{rad}$.

[1]GFZ Helmholtz Centre for Geosciences, Telegrafenberg, Potsdam, Germany. [2]Institute of Geosciences, University of Potsdam, Potsdam, Germany. [3]HUN-REN Institute of Earth Physics and Space Science, Sopron, Hungary. [4]Geology Department, Faculty of Science, Cairo University, Giza, Egypt. ✉e-mail: marzotto@gfz.de; slobanov@gfz.de

One of the reasons as to why radiative heat transport has been largely ignored in geodynamic models is because early studies[23,24] reported that iron-bearing silicates minerals are likely opaque at the high $P$, $T$ conditions of the upper mantle. In order to estimate the contribution of $\Lambda_{rad}$ of representative mantle minerals, it is crucial to probe the optical absorption coefficient in the wavelength range that includes the peak of thermal emission at mantle temperatures $1000 < T < 2500\,K$, i.e., the infrared (IR) range between 1.1 and 2.8 $\mu m$. Several studies have reported estimates of $\Lambda_{rad}$ for relevant iron-bearing silicate minerals: olivine[22-35], pyroxene[27,31,35], garnet[27,31,35], wadsleyite[36], ringwoodite[20,36], bridgmanite[21,37], and ferropericlase[37]. Most of these estimates, however, vary by several orders of magnitude, possibly because measurements were performed on samples of variable composition, optical quality, and at different $P$, $T$ conditions. Quantitative estimates of the radiative component at high $P$, $T$ conditions are very challenging because they require optical measurements on very small samples in the IR spectral range[16,38]. Olivine (Mg,Fe)$_2$SiO$_4$, the most abundant mineral in the slabs at upper mantle conditions, is a case in point. Even at room pressure, the radiative conductivity of olivine is poorly constrained with extant estimates ranging from 0.31[27] to 2.34[22] $W\,m^{-1}\,K^{-1}$ ($P = 1\,atm$, $T \leq 1300\,K$). At room pressure, crystal field bands (d-d excitation of Fe$^{2+}$ atoms) are a prominent absorption feature in olivine in the near-IR range[23,24]. Measurements performed at high-$T$ and room-$P$ revealed that the crystal field bands broaden, and the lattice bands in the mid-IR intensify with increasing temperature[22,25]. Temperature-induced strengthening of the lattice bands is particularly important because these bands may effectively block radiative heat transport at upper mantle temperature. In addition, room-$T$, high-$P$ (6–27 $GPa$) optical experiments on fayalite (i.e., the iron endmember of olivine, Fe$_2$SiO$_4$) revealed that the Fe-O absorption edge shifts into the visible and near-IR range, and becomes the dominant light absorption mechanism, potentially blocking radiative heat transport[23,24]. These studies, however, did not clarify which absorption mechanism(s) govern the optical properties of olivine at upper mantle $P$, $T$

conditions due to the lack of experimental spectral coverage and/or limited $P$, $T$ range[27-30].

In this work, we probed, for the first time, the optical absorption coefficient of olivine ($\alpha^{Ol}$) in the IR and visible spectral ranges at simultaneous high $P$ and $T$. This achievement was enabled by the use of a pulsed white laser probe synchronized to fast spectroscopic equipment in an optical setup for laser-heating diamond anvil cell (LH-DAC) experiments[37-40]. From our $\alpha^{Ol}$ measurements, we derived reliable estimates of the radiative thermal conductivity of olivine ($\Lambda_{rad}^{Ol}$) at upper mantle $P$, $T$ conditions. Our experiments reveal that olivine remains relatively transparent to IR light even at high $P$, $T$, which makes $\Lambda_{rad}^{Ol}$ a non-negligible component of diffusive heat transport in the mantle. Additionally, we derived a new formulation to compute $T$-dependent $\Lambda_{rad}^{Ol}$, which we introduced in a heat diffusion model to investigate the effects of significant radiative heat transport in the thermal evolution of subducting slabs. Recent thermal evolution models of subducting slabs[13,41-49] included $P$,$T$-dependent thermal conductivity formulations, and reported a large temperature difference ($\Delta T = 100$–$200\,K$) compared to the models with constant $\Lambda$. Such large $\Delta T$ has dramatic effects on subduction dynamics[43,44] (i.e., slab buoyancy and sinking velocity), and affects the preservation of hydrous phases inside the slab[49]. In previous subduction models, however, the contribution of $\Lambda_{rad}$ was fixed to small values (<1.5 $W\,m^{-1}\,K^{-1}$), which leads to negligible effects on subduction dynamics[13,41,46,48]. The inclusion of high $\Lambda_{rad}$, such as inferred in this work, might radically change the internal thermal conductivity distribution inside the slab. Our models show that the inclusion of both lattice and radiation heat diffusion mechanisms in geodynamic models is crucial for the correct computation of slab thermal evolution and the estimation of the amount of water released at $< 230\,km$[7,10] or delivered to the MTZ ($\sim$410–660 $km$)[8].

## Results

### The absorption coefficient of olivine at high-$P$, $T$

The absorption coefficient of fayalite $\alpha^{Fa100}$ measured in the LH-DAC at ~1.1 $GPa$ is shown in Fig. 1. Across all frequencies probed, $\alpha^{Fa100}$

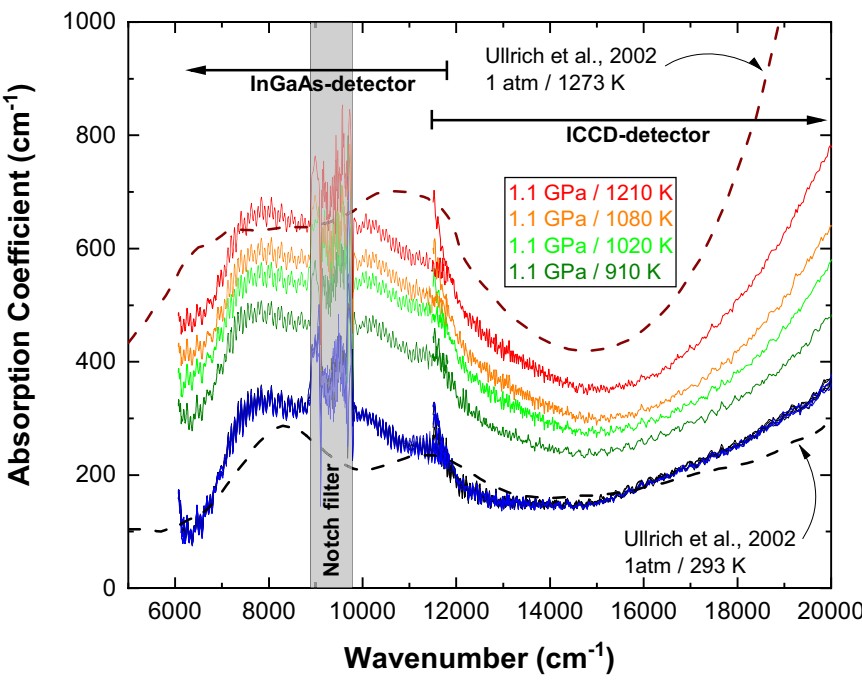

**Fig. 1 | The absorption coefficient of fayalite $\alpha^{Fa100}$ at ~1.1 $GPa$, measured in the laser-heated diamond anvil cell (solid spectra).** Eight overlapping black and blue spectra were measured before and after each of the high-temperature spectra (color-labeled), respectively. The dashed spectra are absorption coefficients of fayalite at room- and high temperature reported by ref. 30. The ICCD and InGaAs labels denote the spectral ranges recorded by these detectors. The vertical grey bar shows the region where our optical measurements are inaccurate due to the notch filter blocking the heating laser.

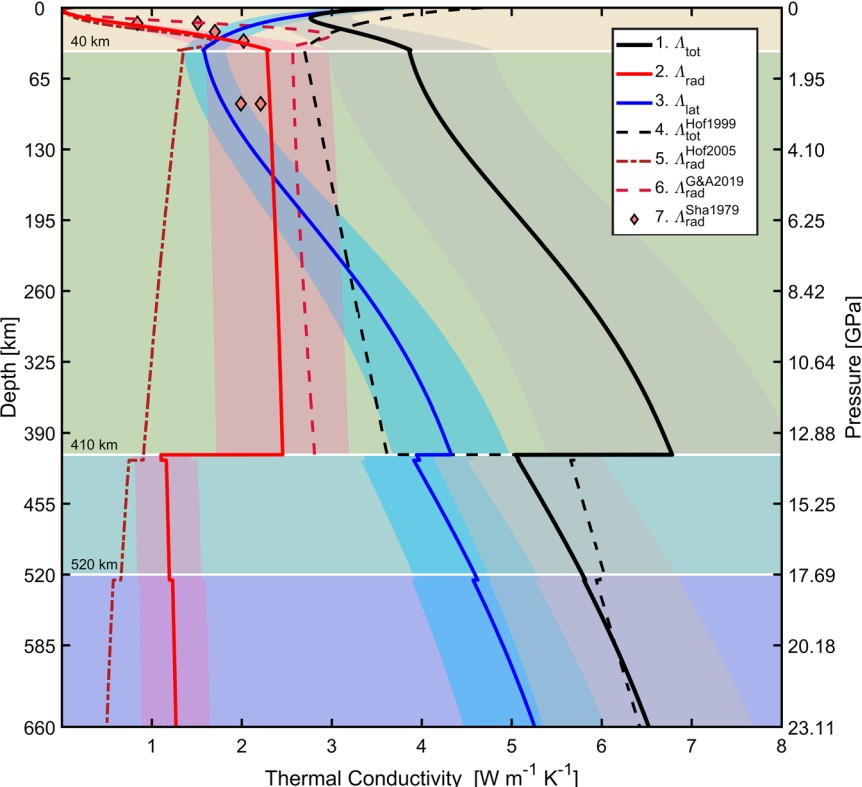

**Fig. 2 | Thermal conductivities of the Earth's outer shell: 40 km-thick lithosphere, upper mantle (40–410 km), upper MTZ (410–520 km), and lower MTZ (520–660 km).** Our calculated mantle adiabat[53,54] assumes a potential temperature of 1619 K. The solid curves represent the $\Lambda$ profiles used in our study, while the dashed and dashed-dotted curves indicate the $\Lambda$ profiles reported in the literature. The $P$, $T$ dependence of $\Lambda_{rad}^{Ol}$ is shown in Fig. S5. 1. (solid black) total thermal conductivity computed as $\Lambda = \Lambda_{lat} + \Lambda_{rad}$. The grey shaded area represents the uncertainty in $\Lambda$ estimates. 2. (solid red) radiative thermal conductivity $\Lambda_{rad}$ computed from: olivine[this study] (UM), wadsleyite[36] (upper MTZ), and ringwoodite[36] (lower MTZ). The red shaded area represents the $\pm 30\%$ uncertainty in $\Lambda_{rad}^{Ol}$ estimates from LH-DAC measurements[37–39]. 3. (solid blue) lattice thermal conductivity $\Lambda_{lat}$ profile computed from: olivine[55,68] (UM) and ringwoodite[45] (MTZ). The blue shaded area represents $\pm 15\%$ uncertainty in $\Lambda_{lat}^{Ol}$ estimates from Time-Domain Thermo-Reflectance (TDTR) measurements[45,68]. 4. (dashed black) total thermal conductivity $\Lambda_{tot}^{Hof1999}$, with $\Lambda_{rad}^{Hof1999} \sim 0.35\ W\ m^{-1}\ K^{-1}$ ref. 27. 5. (dashed-dotted crimson) radiative thermal conductivity $\Lambda_{rad}^{Hof2005}$, ref. 28. 6. (dashed crimson) radiative thermal conductivity $\Lambda_{rad}^{G\&A2019}$, ref. 35. 7. (coral diamonds) radiative thermal conductivity for the crystallographic directions a and c $\Lambda_{rad}^{Sha1979}$, ref. 25. Note that our estimated $\Lambda_{rad}$ (profile 2, solid red) rapidly increases in the lithosphere (linear $T$ gradient 35 K/km) and flattens in the sub-lithospheric mantle (adiabatic $T$ gradient $\sim 0.5\ K/km$[54]).

increases with heating. This increase is fully reversible upon quenching, as indicated by the room-temperature spectra measured after each heating cycle, suggesting no significant irreversible chemical alteration in fayalite at high temperatures. The agreement with $\alpha^{Fa100}$ measured at 1 atm in a furnace[30] is excellent, indicating a homogeneous temperature in the probed laser-heated fayalite. At pressures of ~5 and ~10 GPa, the measured absorption coefficients are very similar in magnitude to those at ~1.1 GPa, albeit the crystal field bands are blue-shifted, consistent with the expected pressure-induced shift[24,26]. The wavelength-dependent temperature-derivatives of $\alpha^{Fa100}$ measured in LH-DACs increase with temperature at all pressures studied here (Fig. S3). Similarly, the absorption coefficient, and its temperature-derivative for natural peridotitic olivine (Fa9.1) determined in the LINKAM experiment, also increase with temperature (Fig. S4). Using representative temperature-derivatives from the LH-DAC, the LINKAM experiments, and the data reported by ref. 25 we obtained a continuous in wavelength temperature-derivative of the absorption coefficient of peridotitic olivine (see Supplementary Materials, SM).

Our results are in agreement with the extant studies of olivine opacity at frequencies > ~4000–5000 $cm^{-1}$ [25,28–30], in particular with the data reported by ref. 25 where they probed $\alpha^{Ol}$ at $P = 1$ atm and $T \leq 1673$ K in the near- and mid- IR spectral range. The agreement with the optical data of ref. 25 indirectly testifies to the reliability of our measurements. In addition, we observe that the absorption coefficient at <4000 $cm^{-1}$ strongly increases with temperature due to the

broadening of the lattice vibration bands, as also reported by ref. 25. This is an important observation that challenges the assumption made by ref. 28, that only the $d$-$d$ crystal field bands contribute to the temperature-induced opacity of olivine at $T > 673$ K. The major differences between our $\alpha^{Ol}$ measurements and most previous room-$P$, high-$T$ studies[28–30] arise from their limited spectral coverage[30], the investigated $T$ range (e.g., $T < 673$ K[28]), or both[29]. At high temperatures, the optical anisotropy of olivine's absorption coefficient $\alpha^{Ol}$ [50–52] results in only ~10% different $\Lambda_{rad}^{Ol}$ [25] (Fig. 2, diamond markers). Therefore, the unpolarized optical measurements reported here enable an adequate assessment of the radiative thermal conductivity of olivine at upper mantle temperatures. We further discuss the anisotropy of $\alpha^{Ol}$ in the SM.

## Olivine's P, T-dependent radiative thermal conductivity

From our absorption coefficient measurements, we computed $\Lambda_{rad}^{Ol}$ (SM) along the $P$, $T$ profile of the upper mantle[53,54]. We find that the radiative conductivity increases steeply below 40 km, but is $\sim 2.3\ W m^{-1} K^{-1}$ throughout the upper mantle (Fig. 2). The relative error of our estimates is $\Lambda_{rad}^{Ol} \pm 30\%$. This is an empirical estimate that takes into account how the uncertainty in thickness and temperature of the sample under high $P$[37–39] propagate to the error in the absorption coefficient. The uncertainty in temperature is particularly important because the absorbance of the sample is measured across a volume subject to a strong temperature gradient along the laser-heating

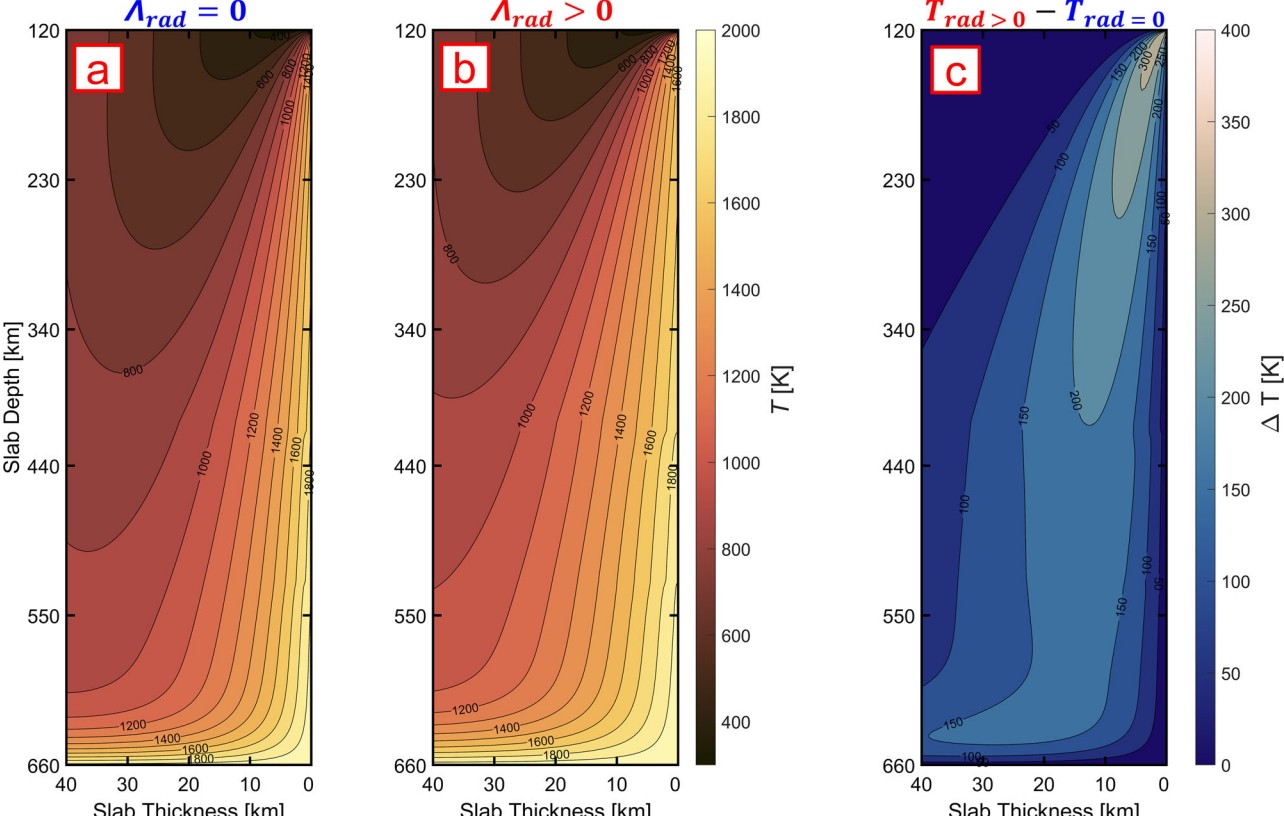

**Fig. 3 | Examples of final 2D temperature distributions from a slab model with sinking velocity of $v_{sink} = 5\ cm\ yr^{-1}$ and age of $t_{slab} = 80\ Myrs$.** Panel (**a**) represents the model without radiative heat transport; panel (**b**) the model with radiative heat transport; panel (**c**) shows the temperature difference between (**b**) and (**a**). Each subplot plot shows only the upper 40 $km$ of the slab, starting from its surface at 0 $km$. For the plot we used the Scientific Color Maps[56]. The temperature profile across the whole slab thickness is reported in Fig. S15.

direction. We find that the radiative component of the total thermal conductivity of olivine $\Lambda^{Ol}$ is almost insensitive to pressure but strongly dependent on temperature (Fig. S5). Moreover, our estimates demonstrate that heat radiation is the predominant mechanism for heat diffusion in the upper part of the sub-lithospheric mantle (<150 $km$), where $\Lambda^{Ol}_{rad} > 0.5\Lambda^{Ol}$. Within the lithosphere (<40 $km$), the lattice component $\Lambda^{Ol}_{lat}$ (Fig. 2) is strongly damped[55] due to the high-$T$, low-$P$ conditions. In the sub-lithospheric mantle, however, the pressure gradient exceeds the temperature gradient, hence $\Lambda^{Ol}_{lat}$ becomes progressively more relevant for $\Lambda^{Ol}$ with increasing depth (Fig. 2). Nevertheless, the high temperatures sustain intensive radiative heat transport down to a depth of 410 $km$, where $\Lambda^{Ol}_{rad} \sim 0.4\Lambda^{Ol}$.

Our radiative conductivity model for olivine is in reasonable agreement with that of ref. 25 but differs substantially from the estimates obtained from other formulations in the literature[27,28,35] (Fig. 2). Please note that the formulation in ref. 35 is based on the optical absorption spectra reported by ref. 28, who assumed that only the $d$-$d$ crystal field bands contribute to the temperature-induced opacity of olivine at $T > 673\ K$, an assumption that is not supported by our LINKAM experiments. As a result, the $T$-derivative of $\Lambda^{Ol}_{rad}$ from ref. 35 is ~2.5 times steeper than that from our work at $T < 1200\ K$. From our estimates, we conclude that olivine's radiative and lattice conductivities are of approximately equal importance at upper mantle conditions.

## Discussion
### The effect of olivine's radiative thermal conductivity on slab thermal evolution

The inclusion of $\Lambda_{rad}$ has a significant effect on the internal temperature distribution of the slab (Fig. 3). Each model shows a cold core

($T < 1000\ K$) surrounded by hot outer edges[56]. However, accounting for radiative transport in heat diffusion (model set 02; $\Lambda_{rad} > 0$) leads to slabs that are~ 100−200 $K$ warmer than in the simulations with only the lattice component (model set 01; $\Lambda_{rad} = 0$). This tendency can be observed in all models (Fig. S14). Using bound values for thermal conductivity (i.e., upper and lower limits of the grey shaded area in Fig. 2) will produce a $\Delta T$ of ± 50 $K$ compared to the reference model in Fig. 3b ($t_{slab} = 80$; $v_{sink} = 5\ cm\ yr^{-1}$; $\Lambda_{rad} > 0$), whereas the formulation from ref. 35 (eq. S12 in SM) produces $\sim$ 20 $K$ hotter slab than the reference in Fig. 3b.

The inclusion of high $\Lambda_{rad}$ formulation in heat diffusion models intensifies the heat flow leading to faster and deeper slab heating compared to $\Lambda_{lat}$-only cases. The $T$-derivatives of $\Lambda^{Ol}_{rad}$ and $\Lambda^{Ol}_{lat}$ have opposite sign: $\left(\frac{\partial \Lambda_{rad}}{\partial T}\right)_P \sim 1.61 \times 10^{-3} W m^{-1} K^{-2}$; $\left(\frac{\partial \Lambda_{lat}}{\partial T}\right)_P \sim -1.03 \times 10^{-3} W m^{-1} K^{-2}$. Therefore, as temperature increases, radiative heat transport is enhanced and overcompensates for the reduction of lattice conductivity induced by $T$ (Fig. 2), which gives a positive $T$-derivatives of total thermal conductivity $\Lambda^{Ol}_{tot}$: $\left(\frac{\partial \Lambda_{tot}}{\partial T}\right)_P \sim 0.58 \times 10^{-3}\ W m^{-1} K^{-2}$. The positive feedback between temperature and $\Lambda_{rad}$ maintains high conductivity within the slab and smooths $\Lambda$ variations across its thickness (Figs. S16, S17). The effect of pressure on $\Lambda_{rad}$, instead, is negligible $\left(\frac{\partial \Lambda_{rad}}{\partial P}\right)_T \sim 0\ W m^{-1} K^{-1} GPa^{-1}$, while the $P$-derivative of lattice conductivity is significant $\left(\frac{\partial \Lambda_{lat}}{\partial P}\right)_T \sim 0.55\ W m^{-1} K^{-1} GPa^{-1}$. Therefore, the variation in $\Lambda^{Ol}_{tot}$ along the slab dip is almost exclusively due to the increasing $\Lambda_{lat}(P)$, hence $\left(\frac{\partial \Lambda_{tot}}{\partial P}\right)_T \sim 0.55\ W m^{-1} K^{-1} GPa^{-1}$. It follows that the horizontal (lateral) variations of $\Lambda$ within a subducting slab are

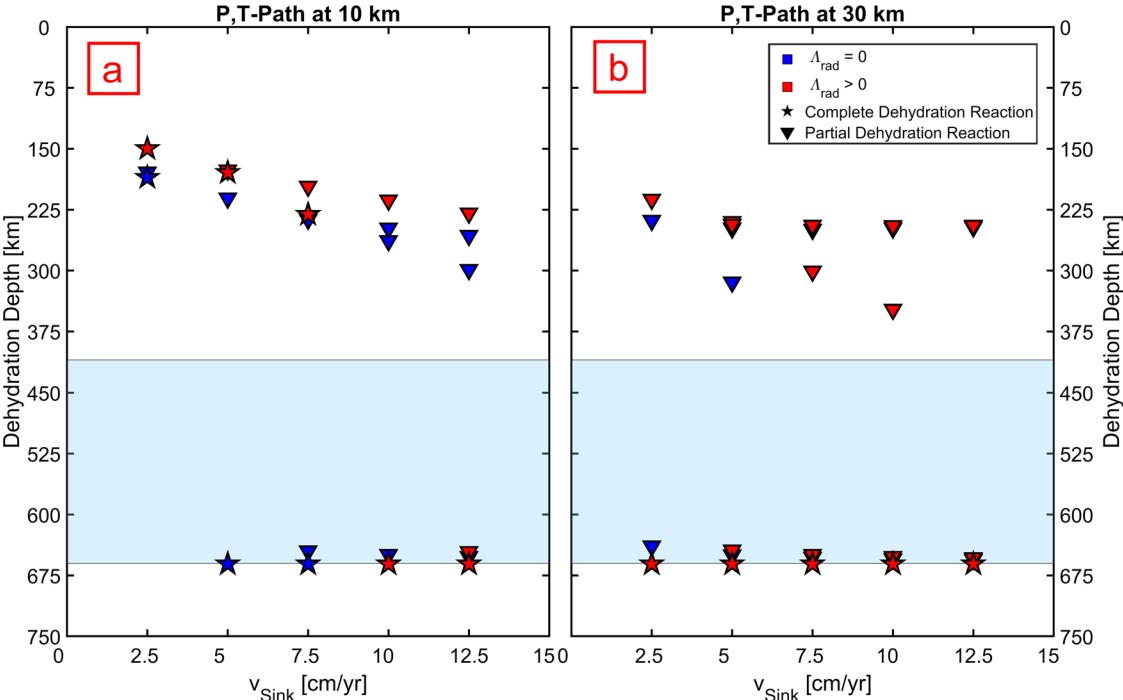

**Fig. 4 | Depth of dehydration reactions as a function of sinking velocity $v_{sink}$.**
Slab age is $t_{slab} = 80\ Myrs$, slab thickness $H_{slab} = 120\ km$. Different markers indicate the type of dehydration reaction (star: complete dehydration; triangle: partial dehydration), whereas the colors indicate the type of model ($\Lambda_{rad} = 0$: blue; $\Lambda_{rad} > 0$: red). The light blue area indicates the MTZ (410–660 $km$). The two subplots indicate different positions of extracted $P, T$ profiles inside the slab: (**a**) 10 $km$ from slab surface (Moho); (**b**) 30 $km$ from slab surface (coldest region within the slab corresponding to maximum hydration depth)[11]. Partial dehydration reactions determine the loss of 50–70% of the initial slab water content at <350 $km$ of depth. The remaining 30% is lost between 640 and 660 $km$ of depth (Fig. S13 in SM).

significantly smaller than the vertical (radial) ones, as also reported by ref. 13. In previous models[13,41,46,48], however, the contribution of $\Lambda_{rad}$ was not sufficient to compensate for the diminishing $\Lambda_{lat}$, and large vertical variations of thermal conductivity were mainly generated by $\Lambda$ jumps across phase boundaries[13]. In our models, in contrast, $\Lambda_{rad}$ contribution attenuates the effects of temperature on slab's total thermal conductivity due to the lattice component, thus making more apparent the effects of pressure given by $\Lambda_{lat}$ contribution.

**Large-scale impact of $\Lambda_{rad}$ on slab dehydration and subduction dynamics**

Previous models[43,44] reported that $P$-dependent $\Lambda_{tot}$ significantly decelerate subduction rate by increasing slab's thermal buoyancy. Potentially, including the contribution of strongly $T$-dependent $\Lambda_{rad}$ into $\Lambda_{tot}$ will dramatically alter the whole slab dynamics. In our models, the largest temperature difference between the two cases ($\Lambda_{rad} = 0$ and $\Lambda_{rad} > 0$) is located in the region extending from the former ocean floor to ∼40 $km$ towards the slab core (Fig. 3c, S15). At the trench, this region is the coldest part of the oceanic lithosphere, experiencing the largest temperature gradient once subduction begins. Moreover, the uppermost part of the slab is characterized by the greatest diversity of lithologies[57,58] and the highest viscosity[59]. Most importantly, this region contains the hydrous minerals formed by seafloor alteration[11].

Hydrous minerals are particularly temperature-sensitive and only survive in the cold regions of the slab[8]. Dehydration is thus triggered at shallower depth in slabs that heat up more rapidly. To quantify slab dehydration, we extracted the final $P, T$ profiles from our models, and tracked the sequence of phase reactions[60] that take place in a simplified hydrous harzburgite mineral assemblage: olivine, enstatite, and antigorite (SM). Importantly, in the $\Lambda_{lat}$-only models, the $P, T$ profiles taken at 10 $km$ from the slab surface (roughly corresponding to the Moho in the oceanic lithosphere) cross only partial dehydration reactions (i.e., the solid reaction products are not all anhydrous),

allowing the slabs to deliver 13–25% of their original water content at the base of MTZ (660 $km$) for all tested subduction velocities. In contrast, models that include radiative heat transport are ∼100–150 $K$ warmer (Fig. S14). This has dramatic consequences: the most superficial part (0–10 $km$) of slow and moderately fast slabs ($v_{sink} < 10cm$ $yr^{-1}$) dehydrate completely before reaching the MTZ (Fig. 4a). Nonetheless, the hydrous minerals can still reach the MTZ even in $\Lambda_{rad}$-including models when they are hosted in the coldest region of the slab, ∼ 30 $km$ below its surface (Fig. 4b). Similar trends are seen in the models of 60–80 $Myrs$ old slabs, whereas 20–40 $Myrs$ old slabs – which are thinner and initially hotter – lose their entire water budget outside of the MTZ (Fig. 4, S18–S20; Extended Data Table S8). Compared to the reference model (Fig. 3b), the 50 $K$ hotter slabs produced with +30% $\Lambda_{rad}$ uncertainty lead to 16 $km$ shallower dehydration (Fig. S13). Similar dehydration depths are obtained when using $\Lambda_{rad}$ formulation from ref. 35. On the other hand, the 50 $K$ colder slab produced with the −30% $\Lambda_{rad}$ uncertainty leads to a 25 $km$ deeper dehydration.

Heating enhanced by the radiative mechanism can have important consequences for the dynamics of the slab, for example, an increase of +100 $K$ due to radiative heating, reduces temperature-dependent viscosity[59] by 2 orders of magnitude, thus favoring plastic deformation in the slab (Figs. S21–S22). Similarly, models that include radiative heat transport are characterized by 15–35 $kg\ m^{-3}$ lower density compared to $\Lambda_{lat}$-only models (Figs. S23–S24), which corresponds to a 0.3–1.0% decrease in total slab density (∼3500 $kg\ m^3$). A + 25 $kg\ m^{-3}$ density contrast is sufficient to maintain subduction[61], therefore a $-35 < \Delta\rho < -15\ kg\ m^{-3}$ contrast should considerably reduce slab descending velocity, as also reported by refs. 43,44. The effects of $\Lambda_{rad}$ on the dynamics of subducting lithosphere warrant detailed future thermo-mechanical modelling.

The experimental breakthrough presented in our work informed a new model of thermal conductivity that properly accounts for the

lattice $\Lambda_{lat}$ and radiative $\Lambda_{lat}$ contributions in the upper mantle. This study demonstrates that olivine is infrared transparent even at the high $P$, $T$ conditions of the upper mantle. It follows that the radiative mechanism significantly contributes to heat diffusion in olivine-rich upper mantle, representing up to ~40% of its total thermal conductivity. For this reason, $\Lambda_{rad}$ plays an important role in slab heating, especially as temperature rises during subduction. Our thermo-kinematic models support this claim and provide solid evidence that $\Lambda_{rad}$ can have far-reaching effects on slab subduction. In particular, we show that the rapid heating enhanced by radiation reduces the breakdown depth of hydrous phases, potentially triggering intermediate-depth seismicity in the slab. In addition, the faster heating might also promote slab bending and stagnation[62] in the MTZ by reducing rock density and viscosity. We therefore suggest that future numerical models of slab subduction should account for radiative heat transport.

## Methods

### Optical measurements in LH-DAC

We probed the temperature-dependence of the optical absorption coefficient of olivine in two types of experiments (SM). In the first, we measured $\alpha^{Ol}$ of non-oriented 350-$\mu m$-thick double polished samples of natural peridotitic olivine (Fa9.1) at 1 $atm$ and up to 1273 $K$ using a high-temperature LINKAM T1500 stage. In the second, we measured $\alpha^{Ol}$ of synthetic non-oriented ~20-$\mu m$-thick single crystals of fayalite (Fa100) in a laser-heated diamond anvil cell (LH-DAC) with an unpolarized broadband (supercontinuum) laser probe (4000–24000 $cm^{-1}$)[37–39]. LH-DAC experiments were carried out at $P \sim 1.1$, $\sim 5$, $\sim 10$ $GPa$ and $T$ up to 1210 $K$. The use of fayalite was necessary for reliable detection of the absorption bands in thin (~15–20 $\mu m$) samples. The absorption coefficient of fayalite, scaled for iron content, shows that fayalite is a suitable model for mantle olivine[63] (Fig. S1). Essential details of Fa100 synthesis and its suitability as a model for mantle olivine are described in the SM. Radiative thermal conductivity of a mantle (Fe-poor) olivine $\Lambda_{rad}^{Ol}$ was then computed using optical data from both types of experiments (SM).

### Numerical model

To quantify the effects of olivine $\Lambda_{rad}^{Ol}$ on the thermal evolution of subducting slabs[13], we performed a series of numerical models using a MATLAB (R2022a)[64] code (SM), that solves the 2D heat diffusion equation with the finite difference method[65]. This code is based on a previously developed algorithm[49]. In our models, we defined a rectangular $L_{slab} \times H_{slab}$ slab, which was discretized into a mesh of square cells, with a constant grid spacing of 500 $m$. Slab length was set to $L_{slab} = 600$ $km$, whereas slab thickness ranged between $60 \leq H_{slab} \leq 120$ $km$ as it was calculated from the slab age: $20 \leq t_{slab} \leq 80$ $Myrs$ (eq. S25). The initial temperature profile of the slab upon entering the mantle was computed with the half-space cooling equation[66] (eq. S26). In the model, the slab subducted vertically (dip angle $\theta_{dip} = 90°$) at a prescribed velocity ($v_{sink}$), while heat flowed from the hot ambient mantle into the cold slab interior as a function of $\Lambda(P, T)$. Subduction occurred from 60–120 $km$ depth (depending on $t_{slab}$) down to 660 $km$, i.e., the base of the MTZ. In the model, we ignored the interactions between the subducting slab and the over-riding plate. During subduction, $P$, $T$ were increased following the PREM[53] and the adiabatic temperature profile of the mantle reported by ref. 54. The $T$-dependent heat diffusion equation is non-linear and requires the re-computation of the thermodynamic properties of the slab using the $P$, $T$ conditions of the current time-step. The equations used to compute $P$, $T$-dependent thermal conductivity $\Lambda_{tot}$ [$Wm^{-1} K^{-1}$], density $\rho$ [$kg\ m^{-3}$], specific heat capacity $Cp$ [$J\ kg^{-1}\ K^{-1}$], and heat diffusivity $\kappa$ [$m^2 s^{-1}$] are reported in the SM (eqs. S6–S9; S13–S24; Extended Data Tables 1, 4). We restricted the mineralogical composition of the slab to olivine and its polymorphic modifications wadsleyite ($Wd$) and ringwoodite ($Rw$), which is adequate because the volumetric

fraction of these minerals is $\sim 60\%$ in the upper mantle[12] and $\sim 80\%$ in the oceanic lithosphere[67]. Therefore $\Lambda^{slab} = \Lambda_{lat}^i(P, T) + \Lambda_{rad}^i(T)$ where $i$ indicates $Ol$ this study[55,68], at shallow depth, $Wd$[36] between 410 and 520 $km$[12], and $Rw$[36,45] between 520 and 660 $km$[12]. The presence of other upper mantle minerals, such as pyroxene or garnets, does not have a strong effect on the radiative thermal conductivity of the whole rock, because the optical properties of these phases are expected to be similar to that of olivine at high temperature[35]. Therefore, the inclusion of olivine and its polymorphs is adequate for estimating the aggregate radiative conductivity of slab lithologies. We also note that, in the computation of $\Lambda_{tot}$, we ignored the effect of grain size (i.e. light and phonon scattering) as they are unimportant to first order (SM). Overall, we ran a total of 40 models divided into two sets: 1) $\Lambda_{rad} = 0$ and 2) $\Lambda_{rad} > 0$. In each set, we varied slab age $t_{slab}$ : [20; 40; 60; 80] $Myrs$ and $v_{sink}$ : [2.5; 5.0; 7.5; 10; 12.5] $cm\ yr^{-1}$. The full description of the physical model and its numerical discretization, the governing equations, the limitations, and the benchmark of the numerical solution are reported in the SM.

## Data availability

The radiative thermal conductivity and the post-processed numerical model data generated in this study are provided in the Supplementary Materials. The figures and tables used in this study can be downloaded from the following the Zenodo repositories: https://doi.org/10.5281/zenodo.15302102 (Figures) https://doi.org/10.5281/zenodo.14288424 (Tables).

## Code availability

The MATLAB, R2022a[64] code developed for numerical calculations is available in the following Zenodo repository: https://doi.org/10.5281/zenodo.14381991.

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

## Acknowledgements

This study has been supported by Helmholtz Young Investigators Group CLEAR (VH-NG–1325). E.M. acknowledges the support of the DFG projects: BL 1690/1-2 and MA 11165/2−1 (project number: 559893073). A.K. thanks funding and support from the Research Grant Hungary (RGH_L 151351) and the International Lithosphere Program. Open Access funding supported by Projekt DEAL. The numerical model has been developed in MATLAB using the license 139702 (GFZ_network_concurrent). E.M. would like to thank M. Jarema for the support in optimizing the MATLAB code, and S. Sarkar for English proof reading. S.B.C. would like to thank Rico Fuchs for his assistance in the gas mixing furnace lab at the University of Potsdam.

## Author contributions

E.M. and S.S.L. conceived and designed the study. S.S.L. conducted the optical LH-DAC experiments, analyzed the absorption spectra and computed the radiative thermal conductivity from the laboratory data. E.M. extrapolated the algorithm to compute radiative thermal conductivity as a function of temperature, designed the numerical model of slab subduction and slab dehydration, and wrote the MATLAB codes. S.P., M.K.-M., A.K. contributed to data interpretation, literature review, and scientific discussion on the large-scale implications. S.B.C. synthetized the fayalite samples, while N. A.-H. polished the samples and prepared them for the LH-DAC experiments. E.M. and S.S.L. wrote the manuscript with input from all authors. All authors reviewed and approved the final version of the manuscript.

## Funding

## Competing interests

The authors declare no competing interests.
