## [Transparent Peer Review file · Nature Communications]

Olivine's high radiative conductivity increases slab temperature by up to 200 K

Corresponding Author: Dr Enrico Marzotto

Version 0:

Reviewer comments:

Reviewer #1

(Remarks to the Author)

In the presented manuscript, Marzotto and coauthors investigate the radiative heat transport in olivine at upper-mantle conditions and its potential effects on the thermal state of subducted oceanic slabs. Based on their experiments, they suggest that radiative heat transport represents up to 40% of the total heat flux in olivine. Using thermal numerical models they show that such increased heat transport produces a temperature increase of up to 200 K in subducted slabs. The paper is well structured and the methods and results are in general well described, although most of the description of the method is in the supplement. The paper is of general interest with a potentially significant impact on models of thermal state and dynamics of subducting slabs. As mentioned by the authors, the influence of radiative heat transport has been mostly neglected in existing subduction models. I hope their results presented here will promote numerical studies of slab dynamics which will fully incorporate spatial variations in physical properties, in particular the thermal conductivity.

As a modeler who mostly runs thermo-mechanical simulations, I cannot judge the quality of the experimental part of the study. However, I have several comments on the presentation of the results. Most importantly, I would like to see a clear statement about the novelty of their approach and results: Why are their results reliable (compared to previous studies)? What is the error estimate? How much do they differ from the results of previous studies? How important are the differences - what effects do they cause in the thermal evolution of slabs? For publication in Nature Communications, I expect that the study presents a real breakthrough - to me it is not clear from the text.

The thermal model, although highly simplified, shows the impact of the radiative heat transfer clearly. I have some problems understanding the setup (see comments below). The dehydration model is quite complex, but the presentation of its results is very basic and should be improved.

In summary, the presented study may importantly change the values of thermal conductivity used in numerical models potentially leading to a better understanding of the evolution of subducting slabs. However, the presentation of the results needs improvement.

Detailed comments:

- Line 43: I think it is not necessary to start the text with Earth habitability when writing about thermal properties of subducting slabs.

- L72: Please consider citing Tosi et al. (2013) and Morishige (2022), who studied the impact of variable thermal properties on the dynamics of subducting slabs and their thermal structures.

Tosi et al. (2013): "Influence of variable thermal expansivity and conductivity on deep subduction." *Subduction Dynamics: From Mantle Flow to Mega Disasters* (2015): 115-133.

Morishige (2022) The thermal structure of subduction zones predicted by plate cooling models with variable thermal properties. *Geophysical Journal International* 229:1490–1502.

- L84-92: As stated, these measurements are very challenging and may have significant errors. What is the estimated error of the presented measurements? Could it be presented graphically in Fig. 2?

- L131: In what respect are the results (in)consistent? What is the possible source for this inconsistency? I think this sentence should be in the next section because it refers to Fig. 2.

- Fig. 2: Is the difference between your result and the result of Grose & Afonso (2019) significant? To me, the main difference seems to be at a depth <40 km (i.e. at low temperature). How much does it affect the results of your thermal models?

- L175: You have flat 410- and 660-km transitions. If the depth of the phase transitions varied with temperature, how would it affect slab dehydration?

- L187, Fig. 4:

How do you define partial dehydration?

The figure is not very informative. Suggestion: Instead of the triangles and stars, can you for each model draw a line that will show the evolution of the water content in the slab (at 10 km or 30 km from the slab surface)?

- L197-205: This paragraph is not very well organized. A few introductory words would help.

- L200 "For comparison..." - It is not relevant to mention subduction initiation here as you only model subducting slabs deep in the mantle.

Supplement:

- L291: What is the effect of the constant diffusivity here (and in equation S13, I assume)?

- L295: "which approximately correspond to "?

- Fig. S7: I find this figure misleading - perhaps an outline of the actual model domain would help to understand what you want to show.

- L341: Is the P-T-dependence of density and heat capacity important for your results?

- L344: I think all relevant information should be present in this supplement, not referring to a supplement of another paper. I would especially appreciate a list of model limitations here (or even in the main text). The reference should be to [46], correct?

- L375: depth higher then -> depth greater than

- L380: Below 120 km of depth -> At depths shallower than 120 km

- L381: There is no trench in the model

- Fig. S9: There is no ambient mantle at the top boundary.

- Fig. S17c: Please add -1 contour

(Remarks on code availability)

Reviewer #2

(Remarks to the Author)

Dear Editor,

Myself and the co-reviewer have carefully reviewed the manuscript by Marzotto et al. entitled "Olivine's high radiative conductivity increases slab temperature by up to 200 K". Overall, the work is well-written, logically structured, and the scientific content is of high quality and interest. The research question is clearly stated, the methodology is sound, and the results were tested extensively and are thoroughly discussed within the context of existing literature.

The introduction provides an excellent overview of the topic, clearly highlighting the importance of understanding slab thermal evolution and its implications for deep Earth processes like water transport and plate tectonics dynamics. The experimental methods used to measure the absorption coefficient of olivine at high P-T conditions are state-of-the-art and well described (including the explanation about the use of Fa100 representing Mg-richer compositions). The use of both the laser-heated diamond anvil cell and furnace experiments adds robustness to the results. We can say that the calculations of olivine's radiative thermal conductivity from the absorption coefficient data are rigorous and well-justified. The numerical modeling approach to quantify the effect of radiative heat transport on slab thermal evolution is appropriate and well-implemented. Overall, this is an excellent piece of research that advances our understanding of the role of radiative heat transport in the thermal evolution of subducting slabs and its potential impact on deep Earth processes. The authors demonstrate convincingly that at upper mantle conditions, up to 40% of the total thermal conductivity of olivine is due to radiative (as opposed to lattice) heat transfer. The final result (a likely overlooked higher slab temperature up to 100-200 K) is significant and has implications for water transport to the MTZ. Moreover, this work will allow new and important insights to

be made about the behaviour of slab at the MTZ due to, previously underestimated, slab densities and viscosities.

We have no major criticisms or concerns about the scientific content or presentation of the work. We would recommend publication with only very minor revisions, which we report here below, in some cases these comments are simply “food for thought” and need not be addressed in the revised version of the manuscript:

Lines 68-69: For the benefit of the reader, it may be worthwhile to expand slightly more the background context of this work; i.e., earlier work (e.g., Mao, 1976 and others) suggested that Fe-bearing mantle minerals become opaque at high pressure and thus it was assumed that heat transfer via radiation was negligible. Some years later it was shown this is not necessarily the case, and it may be useful to explain P/T induced mechanisms that result in opacity/transparency changes in olivine and its polymorphs (rather than just simply saying the opacity, and thus degree of radiative heat transfer, is a function of P, T).

Line 97-98: From this section, it is not clear at which pressure this experiment was done, while it is specified the temperature at which the optical absorption was carried out.

Line 98: Would be interesting to compare absorption coefficients obtained from a series of LH-DAC experiments on oriented crystals using polarized light, we assume averaged results would be in agreement with those reported here but would be interesting to compare the oriented results nonetheless.

Line 106: We wonder how difficult it would be to translate all the work done in this study from 2D to 3D (i.e., model thermal diffusivity in 3D and apply to rectangular slab model), and whether this would affect the final conclusions.

Lines 110-113: These assumptions about the modal abundance of olivine in the slab are well-justified, however, is it reasonable to then assume predictions about the slab temperature increase represents a sort of maximum estimate? For example, in a slab with only 60% olivine (instead of 100%) the increase in temperature due to consideration of Δrad (reported as 100-200K by the authors) may be significantly less if the Δrad contribution by the other 40% of the slab is relatively lower.

Line 148: Although accessible from the data, it would be very useful to the reader to provide an in-text expression for $\Delta rad Ol \sim X \Delta Ol$ values such that the reader can determine X for a series of different depths, e.g. X = 0.4 for a depth of 410km.

Lines 203-205: This sentence about how the radiative heat transport may promote slab bending and stagnation can probably be removed as it is repeated again on lines 214-215.

- For the reference n. 40 (MATLAB) it could be useful to write the version of MATBLAB used in the main text.

- Not clear why for reference n. 41 in the reference list, the authors added the doi number, which is missing for all other references.

- Please have a look at the format for all references in the reference list, sometimes we see the issue number, sometimes this number is not reported.

(Remarks on code availability)

Reviewer #3

(Remarks to the Author)

(Remarks on code availability)

Version 1:

Reviewer comments:

Reviewer #1

(Remarks to the Author)

The authors did a good job in addressing my concerns, especially in extending the Supplementary material with more detailed descriptions of the model setup and its limitations. I have only a few additional comments:

1. I think that the error estimate of the presented experimentally-derived radiative conductivity still needs a bit more explanation. As a reader, I would like to have some insight into why 30% is the right value (and not, e.g., 3% or 50%). Were there other studies with a similar technique that came to this value? Or does it come from your experience with the experimental setup? I understand that it is difficult to rigorously calculate the error, but I suppose there is some standard way

to deal with that.

2. To include your T-dependent radiative conductivity in a subduction model, I would ideally have to follow the procedure described in the Supplement (S6). It is possible, but not straightforward, compared to, for example, analytical formulas provided by other authors. Could you suggest a more straightforward way to use your data?

It seems to me that a linear interpolation between the data points in the Extended Data Table 1 will work fine in many applications. An explicit reference to this table may be useful in the main text. It would give a reader a hint on how to use your results without reading the entire supplement. It may increase the impact of your work!

3. I attach a pdf of the manuscript with a few minor corrections and suggestions that you may consider.

(Remarks on code availability)

Reviewer #2

(Remarks to the Author)

I am happy with the revisions carried out by the authors. In my opinion, they addressed all concerns/questions/suggestions by the reviewers. I don't request further changes.

(Remarks on code availability)

We thank the three reviewers for the encouraging feedback and constructive suggestions that helped us prepare our manuscript for resubmission. As is documented below, we have fully addressed all the reviewers' concerns and questions by providing additional details with the resubmitted manuscript. Below, all reviewers' comments are in italics and our point-by-point responses are in normal fonts with blue colour. Changes in our revised manuscript are also labelled by blue colour.

Reviewer #1 (Remarks to the Author):

In the presented manuscript, Marzotto and co-authors investigate the radiative heat transport in olivine at upper-mantle conditions and its potential effects on the thermal state of subducted oceanic slabs. Based on their experiments, they suggest that radiative heat transport represents up to 40% of the total heat flux in olivine. Using thermal numerical models, they show that such increased heat transport produces a temperature increase of up to 200 K in subducted slabs. The paper is well structured, and the methods and results are in general well described, although most of the description of the method is in the supplement. The paper is of general interest with a potentially significant impact on models of thermal state and dynamics of subducting slabs. As mentioned by the authors, the influence of radiative heat transport has been mostly neglected in existing subduction models. I hope their results presented here will promote numerical studies of slab dynamics which will fully incorporate spatial variations in physical properties, in particular the thermal conductivity.

We thank Reviewer #1 for recognizing the significance of our work.

1. As a modeler who mostly runs thermo-mechanical simulations, I cannot judge the quality of the experimental part of the study. However, I have several comments on the presentation of the results. Most importantly, I would like to see a clear statement about the novelty of their approach and results: Why are their results reliable (compared to previous studies)? What is the error estimate? How much do they differ from the results of previous studies? How important are the differences - what effects do they cause in the thermal evolution of slabs? For publication in Nature Communications, I expect that the study presents a real breakthrough - to me it is not clear from the text.

We thank Reviewer #1 for providing this clear and structured comment. We have organized our reply accordingly by addressing each question separately.

1. NOVELTY. The primary novelty is that we report on the optical opacity of olivine at simultaneous high pressure- high temperature conditions representative of the upper mantle, which has never been probed before. This achievement is enabled by the use of a pulsed white laser probe synchronized to fast spectroscopic equipment in an optical setup for laser-heating diamond anvil cell experiments (Lobanov *et al.*, *Rev. Sci. Instrum.*, 2020). Our high-pressure experiments at ~1.1, ~5.0, and ~10.0 GPa revealed that pressure has a minor effect on the temperature-induced opacity of olivine, and thus on its radiative conductivity (Fig. S5). In addition to the use of a new experimental approach, our work is novel in the introduction of

high radiative conductivity in thermal evolution models of slab subduction. We discussed the novelty of our work in Lines 99-108 of the main manuscript.

2. RELIABILITY. The results of our optical measurements are in agreement with those of *Shankland et al. (1979)*, who probed, the opacity of olivine at **1 atm** to 1673 K in the spectral range up to 6000 nm. The latter is particularly critical for quantifying the radiative conductivity at upper mantle temperatures (1000-2000 K) with the peak of the black body emission ranging from ~2900 nm (at 1000 K) to ~1450 nm (at 2000 K). The agreement with the optical data of *Shankland et al., (1979)* indirectly testifies to the reliability of our radiative conductivity model. We discussed the reliability of our work in Lines 187-194 of the main manuscript.

3. ERROR ESTIMATE. We estimated the total relative error to be ~ 30%. This propagated error is primarily sensitive to the uncertainty in sample thickness at high pressure and, to a smaller extent, to the uncertainty on temperature (*Lobanov et al., Earth Planet. Sci. Lett., 2020; 2021*). As requested, we show this error in the revised version of Figures 2 and S5, and we have added a new section of the supplementary materials called 'Uncertainties in radiative thermal conductivity estimates and its anisotropy' at Line 184. We discussed the error estimates also in Lines 206-208 of the main manuscript.

4. COMPARISON TO PREVIOUS STUDIES. Our model of olivine radiative thermal conductivity differs substantially from the model of *Grose & Afonso (2019)*, especially at depths < 40 km (Fig 2). Please note that the model of *Grose & Afonso (2019)* is based on the optical absorption spectra reported by other groups. These previous studies of olivine opacity at high temperature and at **1 atm** either did not provide enough spectral coverage (*Ullrich et al., Phys. Chem. Min., 2002*), were limited to $T < 673$ K (*Hofmeister, J. Geodyn., 2005*), or both (*Taran & Langer, Phys. Chem. Min., 2001*). The olivine radiative conductivity model presented by *Grose & Afonso (2019)* inherits these shortcomings. The comparison between our experiments and literature studies is discussed in the main manuscript at Lines 194-196 (absorption coefficient measurements) and Lines 218-224 (radiative thermal conductivity estimates).

Additionally, our slab thermal evolution model is different from previous numerical models because they either: (a) assumed constant thermal conductivity, $\lambda_{tot} = k$ (e.g. *van Keken et al., 2008; Syracuse et al., 2010*); (b) neglected completely the contribution of the radiative component, $\lambda_{rad} = 0$ (e.g., *Maierova et al., 2012; Tosi et al., 2013,2015*); or (c) considered radiative contribution to be small, as resulting from *Hofmeister (1999, 2005)* formulations:

$\lambda_{rad}^{no\ 1999} \sim 0.35\ W\ m^{-1}\ K^{-1}$; $\lambda_{rad}^{no\ 2005} < 2\ W\ m^{-1}\ K^{-1}$ (e.g., *Hauck et al., 1999; Brandlund et al, 2000; Morishige, GJI, 2022; van Zelst et al, 2023*). In our work, instead, we included a new formulation of radiative thermal conductivity which gives $\lambda_{rad} > 2\ W\ m^{-1}\ K^{-1}$ at upper mantle conditions.

5. IMPLICATIONS FOR THERMAL EVOLUTION OF SUBDUCTING SLABS. The main implication of our work is that geodynamic models should include the radiative component of thermal conductivity when computing diffusive heat transport in the upper mantle. As reported in our study, the T -derivatives of olivine's radiative conductivity has opposite sign and overcompensate the T -derivative of the lattice contribution:

$(\frac{\partial \lambda_{rad}}{\partial T})_p \sim 1.61 \times 10^{-3}\ W\ m^{-1}\ K^{-2}$; $(\frac{\partial \lambda_{at}}{\partial T})_p \sim 1.03 \times 10^{-3}\ W\ m^{-1}\ K^{-2}$. The P -derivative of radiative conductivity, instead, is negligible $(\frac{\partial \lambda_{rad}}{\partial P})_T \sim 0\ W\ m^{-1}\ K^{-1}\ GPa^{-1}$ while the P -

derivative of lattice conductivity is high ($\kappa_{lat} \sim 0.55 P^{-1} T^{-1}$). Therefore, the resulting olivine's total thermal conductivity in a descending slab will be predominantly P -dependent ($\kappa_{tot} \sim 0.55 P^{-1} T^{-1}$), and mildly T -dependent ($\kappa_{rad} \sim 0.58 \times 10^{-3} P^{-1} T^{-2}$). Pressure(depth)-dependent A_{tot} significantly reduces slab's sinking velocity by increasing its thermal buoyancy (Tosi *et al.*, 2013,2015). Therefore, including the contribution of strongly T -dependent A_{rad} into A_{tot} will dramatically alter the whole slab dynamics. In addition, our models show that the constant thermal conductivity assumption can only be reasonable at shallow depths (0-150 km), where the P , T are low and the sum of A_{lat} and A_{rad} is sufficiently small to be similar to the conventionally chosen constant values, e.g. $2.5\text{-}3 \text{ W m}^{-1} \text{ K}^{-1}$. We discussed the effects of radiative thermal conductivity on the thermal evolution of subducting slabs in Lines 238-248 of the main manuscript.

6. FINAL REMARKS: To summarize, the experimental breakthrough presented in our work informed a new model of depth-dependent thermal conductivity that properly accounts for the lattice and radiative contributions in the upper mantle. Moreover, by including the strong T -dependent formulation of A_{rad} , our numerical models present a substantial difference from previous studies, while also providing further insight into heat diffusion in mantle rocks. The implications resulting from the use of the new conductivity formulation are significant and are of broad interest to the geoscience community.

2. *The thermal model, although highly simplified, shows the impact of the radiative heat transfer clearly. I have some problems understanding the setup (see comments below). The dehydration model is quite complex, but the presentation of its results is very basic and should be improved.*

We agree with Reviewer #1, and we have extended the description of the model setup in the main text (see Lines 137-169), and added 2 new figures (Fig. S12-S13) in the supplementary material that better represents the results of our dehydration model. Moreover, we added a further description of the results in the caption of Fig. S18-S20.

3. *In summary, the presented study may importantly change the values of thermal conductivity used in numerical models potentially leading to a better understanding of the evolution of subducting slabs. However, the presentation of the results needs improvement.*

We agree with Reviewer #1, and we have improved the presentation of the model results: Fig. 4 of the revised manuscript, Fig. S13, S18-S20 of the supplementary material.

Detailed comments:

4. *Line 43: I think it is not necessary to start the text with Earth habitability when writing about thermal properties of subducting slabs.*

We accept Reviewer #1 suggestion, and we have removed the reference to Earth's habitability. The revised first few sentences (Lines 44-45) now focus only on plate tectonics and reads:

“Earth is the only known planet featuring plate tectonics^[1], whose main driving force is the gravitational pull of the oceanic lithosphere sinking into the mantle, namely: slab subduction^[2].”

5. L72: Please consider citing Tosi et al. (2013) and Morishige (2022), who studied the impact of variable thermal properties on the dynamics of subducting slabs and their thermal structures. Tosi et al. (2013): "Influence of variable thermal expansivity and conductivity on deep subduction." *Subduction Dynamics: From Mantle Flow to Mega Disasters* (2015): 115-133. Morishige (2022) *The thermal structure of subduction zones predicted by plate cooling models with variable thermal properties. Geophysical Journal International* 229:1490–1502.

We thank Reviewer #1 for suggesting these papers, they are indeed relevant for our work, and they fit with the sentence in Line 72 (now Lines 108-111) of the manuscript. We added Tosi et al. (2013) as Ref.^[43]; Tosi et al. (2015) as Ref.^[44] and Morishige (2022) as Ref.^[46].

6. L84-92: As stated, these measurements are very challenging and may have significant errors. What is the estimated error of the presented measurements? Could it be presented graphically in Fig. 2?

We estimate the overall error in radiative conductivity of olivine at upper mantle conditions at ~30% relative, which is a propagated error based primarily on the uncertainty in sample thickness at high pressure and to a smaller extent to the uncertainty in temperature (Lobanov et al., *Earth Planet. Sci. Lett.*, 2020; 2021). We thank Reviewer #1 for this suggestion and gladly add this error assessment to Fig. 2 and Fig. S5.

7. L131: In what respect are the results (in)consistent? What is the possible source for this inconsistency? I think this sentence should be in the next section because it refers to Fig. 2.

We modified the text at Lines 190-196 in order to clarify the comparison to previous studies and to highlight the source of discrepancy with the results of Hofmeister, 2005 (now Ref.^[28]). We also added another reference: Taran & Langer, *Phys. Chem. Min.* 2001 (now Ref.^[29]).

8. Fig. 2: Is the difference between your result and the result of Grose & Afonso (2019) significant? To me, the main difference seems to be at a depth <40 km (i.e., at low temperature). How much does it affect the results of your thermal models?

We agree with Reviewer #1 that our new radiative conductivity model and that of Grose & Afonso (2019) are significantly different at depth < 40 km (Fig.2), because our data shows that lattice vibration also contribute to the increase in the absorption coefficient, which results in a lower value of A_{rad} at < ~40 km. Compared to ours, the T -derivative of Grose & Afonso (2019) formulation at $T < 1200$ K is ~2.5 times steeper, which leads to faster slab heating at shallow depths. Using the Grose & Afonso (2019) formulation in a thermal evolution model produces a ~50 K hotter slab at a depth shallower than 300. The 50 K hotter temperatures lead to 16 km shallower slab dehydration or Grose & Afonso (2019) model, compared to the

reference model [80 Myrs; 5 cm yr⁻¹; $A_{rad} > 0$] shown in Fig. 3.b (164 km vs 180 km). We discussed the differences between our formulation and the one proposed by Grose & Afonso in Lines 218-224; Lines 233-237 and Lines 285-289 of the main manuscript, and in Fig. S13 of the supplementary materials

9. *L175-: You have flat 410- and 660-km transitions. If the depth of the phase transitions varied with temperature, how would it affect slab dehydration?*

Our dehydration model includes the reactions involving the presence of hydrous wadsleyite (~410-km discontinuity) and hydrous ringwoodite (~660-km discontinuity), as it is based on Ref.^[66] (Komabayashi et al., JGR, 2004). Therefore, in our dehydration model, the effects of a T -dependent transition zone depth are already included, and these results are reported in the original Fig. 4 of the main manuscript. The T -induced deflection of the 410- km discontinuity is reported in the Extended Data Table 12 in the SM. We thank Reviewer #1 for pointing out that this might not be clear to the reader. For this reason, we decided to add a new Figure S13 to the Supplementary Materials, which highlights the T -dependent transition zone (see blue area). Concerning the flat 410- 660- discontinuities in the thermal model, our thermal conductivity model presents a drop in thermal conductivity at 410 km (Fig. 2 in the main manuscript). This drop in thermal conductivity does not affect our conclusions because most of the dehydration reactions occur at a depth shallower than 350 km (Fig. 4 in the main manuscript).

10. *L187, Fig. 4: How do you define partial dehydration? The figure is not very informative. Suggestion: Instead of the triangles and stars, can you for each model draw a line that will show the evolution of the water content in the slab (at 10 km or 30 km from the slab surface)?*

We thank Reviewer #1 for pointing out that our partial dehydration definition is not clear. The operative definition of partial dehydration used in our study can be summarized as: “when the pressure-temperature subduction path of a system crosses a reaction boundary, and the solid reaction products are not all anhydrous, the system is subject to a partial dehydration”. We have added this definition in the description of the dehydration models in Fig. S13 of the Supplementary Materials.

We thank Reviewer #1 for suggesting the use of a line plot for the relative water content inside the slab. However, we believe that the resulting plot (see Figure below) does not provide clearer information to the reader. The figure is a descending step plot from 100% to 0%. The initial water content is preserved until the slab encounters a dehydration reaction (defined as a reaction that releases H₂O), after which it suddenly decreases by some percentage, thus creating the step. We decided to limit the plotted lines to the two A_{lat} -only and A_{rad} -including scenarios for each slab model, and we separated the results into independent figures. If plotted in a single figure, the dehydration path of the different models would overlap with each other, and the plot would be especially busy at < 300 km, where most dehydration reaction occurs. Given their complexity, these convoluted plots are commonly called “Tokyo subway maps”. The figure suggested by Reviewer #1 has the advantage of showing the evolution of water content during slab subduction. However, compared to Fig. 4 of the original manuscript, it does not simplify to the reader the localization of specific dehydration depths. Additionally, this type of plot

increases from 4 to 20 the number of figures necessary to visualize the effects of A_{rad} on slab dehydration. This information is conveniently summarized by Fig. 4 of the main manuscript and Fig. S18-S20 of the supplementary material. For these reasons, we prefer to keep the original Figures without including the new step dehydration plot. Nevertheless, we agree with Reviewer #1, and we decided to report the percentage of water loss in the caption of Fig. 4. Moreover, as reported in the Supplementary Materials, Lines 603-604: “The final list of reactions, and the evolution of slab’s water content, of each model is reported in the excel files attached to this paper.”

11. L197-205: This paragraph is not very well organized. A few introductory words would help.

We thank Reviewer #1 for pointing out that this paragraph (now Lines 291-300) of the Discussion can be improved. We have added a brief introduction to this paragraph: “*Heating*

enhanced by the radiative mechanism can have important consequences for the dynamics of the slab, for example, an increase of +100 K due to radiative heating, reduces temperature-dependent viscosity^[65] by 2 orders of magnitude, thus favoring plastic deformation in the slab (Figs. S21-S22)".

12. L200 "For comparison..." - It is not relevant to mention subduction initiation here as you only model subducting slabs deep in the mantle.

We agree with Reviewer #1 and we have reformulated the sentence as: "A $+25 \text{ k}^{-3}$ density contrast is sufficient to maintain subduction^[67], therefore a $-15-35 \text{ k}^{-3}$ contrast should considerably reduce slab descending velocity, as also reported by Ref.^[43,44]"

Supplement:

13. L291: What is the effect of the constant diffusivity here (and in equation S13, I assume)?

The assumption of a constant heat diffusivity value in equations S25 and S26 is justified by the necessity to create homogenous slab thicknesses H_{slab} and initial temperature T_x profiles for all models, to assess the influence of olivine's radiative thermal conductivity in the thermal evolution of subducting slabs. These conditions were equal for both sets of models (with and without A_{rad}), hence they did not affect the outcome of our simulations. In the Supplementary Material we discussed the limitations of equations S25 and S26, see Lines 360-362. We added a discussion about the effects of constant diffusivity in Lines 372-376, of the supplementary material and Fig. S7.

14. L295: "which approximately correspond to"?

We accept reviewer #1 input, and we added the word "approximately" in Line 295 (now Line 363 of the supplementary material).

15. Fig. S7: I find this figure misleading - perhaps an outline of the actual model domain would help to understand what you want to show.

We acknowledge Reviewer #1 comment, and we have substituted Fig. S7 (now Fig. S9). In the new figure we report the model domains at different time frames of the subduction, which fits better with the slightly modified caption (see red text):

"Representation of slab subduction in our models. The temperature field is plotted using the Scientific Color Maps^[51]. The cold slab (dim colored rectangle) subducts vertically ($= 90^\circ$) – with constant sinking velocity k – inside the hot mantle (bright coloured box). In the model we can prescribe a maximum depth of subduction, which we set to 660 k for this study. Each subplot shows a different timeframe of slab's thermal evolution. Note that this is just a simplified scheme of slab subduction: our model is purely thermal, and we computed the mantle just as a boundary condition: i.e., as one extra stencil of nodes outside of the slab (see Fig. S11)."

16. L341: *Is the P-T-dependence of density and heat capacity important for your results?*

Density (ρ) and heat capacity (C_p) have a strong pressure and temperature dependencies, respectively. It was thus important for us to incorporate their P,T -dependence for the evaluation of the depth-dependent thermal diffusivity $\kappa = \lambda/(\rho * C_p)$.

17. L344: *I think all relevant information should be present in this supplement, not referring to a supplement of another paper. I would especially appreciate a list of model limitations here (or even in the main text). The reference should be to [46], correct?*

We agree with Reviewer #1, and we provide a detailed description of our model Supplementary Materials: calculation of density and specific heat capacity (Lines 320-350, Fig S7); model limitations (Lines 440-513); model benchmark (Lines 515-523). We also thank the Reviewer for correcting the reference and apologize for confusion.

18. L375: *depth higher then -> depth greater than*

We accept this suggestion (Line 375, now Line 434).

19. L380: *Below 120 km of depth -> At depths shallower than 120*

km We accept this suggestion (Line 380, now Line 429).

20. L381: *There is no trench in the model*

We clarified our original statement (line 381, now Line 434-436) with the following: “At depths shallower than 120 km the boundaries were designed to maintain the slab at the initial 1D temperature profile τ , see equation (S26)”.

21. Fig. S9: *There is no ambient mantle at the top boundary.*

We agree with Reviewer #1, and we have modified Fig. S9 (now Fig. S11) by adding clearer labels.

22. Fig. S17c: *Please add -1 contour*

We added the -1 contour in Fig. S17.c (now Fig. S21.c) as requested by Reviewer #1.

Reviewer #2 (Remarks to the Author):

Dear Editor,

Myself and the co-reviewer have carefully reviewed the manuscript by Marzotto et al. entitled "Olivine's high radiative conductivity increases slab temperature by up to 200 K". Overall, the work is well-written, logically structured, and the scientific content is of high quality and interest. The research question is clearly stated, the methodology is sound, and the results were tested extensively and are thoroughly discussed within the context of existing literature.

The introduction provides an excellent overview of the topic, clearly highlighting the importance of understanding slab thermal evolution and its implications for deep Earth processes like water transport and plate tectonics dynamics. The experimental methods used to measure the absorption coefficient of olivine at high P-T conditions are state-of-the-art and well described (including the explanation about the use of Fa100 representing Mg-richer compositions). The use of both the laser-heated diamond anvil cell and furnace experiments adds robustness to the results. We can say that the calculations of olivine's radiative thermal conductivity from the absorption coefficient data are rigorous and well-justified. The numerical modelling approach to quantify the effect of radiative heat transport on slab thermal evolution is appropriate and well-implemented. Overall, this is an excellent piece of research that advances our understanding of the role of radiative heat transport in the thermal evolution of subducting slabs and its potential impact on deep Earth processes. The authors demonstrate convincingly that at upper mantle conditions, up to 40% of the total thermal conductivity of olivine is due to radiative (as opposed to lattice) heat transfer. The final result (a likely overlooked higher slab temperature up to 100-200 K) is significant and has implications for water transport to the MTZ. Moreover, this work will allow new and important insights to be made about the behaviour of slab at the MTZ due to, previously underestimated, slab densities and viscosities.

We have no major criticisms or concerns about the scientific content or presentation of the work. We would recommend publication with only very minor revisions, which we report here below, in some cases these comments are simply "food for thought" and need not be addressed in the revised version of the manuscript:

We thank Reviewers #2 and #3 for such a high praise of our work and their suggestions.

1. *Lines 68-69: For the benefit of the reader, it may be worthwhile to expand slightly more the background context of this work, i.e., earlier work (e.g., Mao, 1976 and others) suggested that Fe-bearing mantle minerals become opaque at high pressure and thus it was assumed that heat transfer via radiation was negligible. Some years later it was shown this is not necessarily the case, and it may be useful to explain P/T induced mechanisms that result in opacity/transparency changes in olivine and its polymorphs (rather than just simply saying the opacity, and thus degree of radiative heat transfer, is a function of P, T).*

We thank Reviewer #2 for this suggestion and expand gladly on the earlier work and relevant light absorption mechanisms, see Lines 73-78 in the main manuscript:

“One of the reasons as to why radiative heat transport has been largely ignored in geodynamic models because early studies^[23,24] reported that iron-bearing silicates minerals are likely opaque at the high P, T conditions of the upper mantle. In order to estimate the contribution of A_{rad} of representative mantle minerals, it is crucial to probe the light absorption coefficient in the wavelength range that includes the peak of thermal emission at mantle temperatures $1000 < T < 2500$ K i.e., the infrared (IR) range between 1.1 - 2.8 μm .”

And lines Lines 87-98 in the main manuscript:

“At room pressure, crystal field bands (d-d excitation on Fe^{2+} atoms) are a prominent absorption feature in olivine in the near-IR range^[23,24]. High-T, room-P measurements performed revealed that the crystal field bands broaden, and the lattice bands, in the mid-IR, intensify with increasing temperature^[22,25]. The temperature-induced strengthening of the lattice bands is particularly important because these bands may effectively block radiative heat transport at upper mantle temperature. In addition, room-T, high-P (6-27 GPa) optical experiments on fayalite (i.e., the iron endmember of olivine, Fe_2SiO_4) revealed that the Fe-O absorption edge shifts into the visible and near-IR range, and becomes the dominant light absorption mechanism, potentially blocking radiative heat transport^[23,24]. These studies, however, did not clarify which absorption mechanism(s) govern the optical properties of olivine at upper mantle P-T conditions due to the lack of experimental spectral coverage and/or limited P,T range^[27-30].”

2. *Line 97-98: From this section, it is not clear at which pressure this experiment was done, while it is specified the temperature at which the optical absorption was carried out.*

We thank Reviewer #2 for bringing this up. We have now specified the pressure at which we performed the laser-heating experiments (Line 129-130 of the main manuscript).

3. *Line 98: Would be interesting to compare absorption coefficients obtained from a series of LH-DAC experiments on oriented crystals using polarized light, we assume averaged results would be in agreement with those reported here but would be interesting to compare the oriented results, nonetheless.*

We thank Reviewer #2 for this very reasonable comment. Indeed, orthorhombic olivine is optically anisotropic (see newly added references 59-61 from the ARIA database <https://eodg.atm.ox.ac.uk/ARIA/data?Minerals/Olivine>). Its radiative thermal conductivity is thus direction-dependent. We expect, however, that at upper mantle temperatures the thermal broadening of crystal field and lattice band diminishes the anisotropy of the absorption coefficient. This inference is supported by *Shankland et al (1979)* who evaluated the radiative conductivity along a and c directions (strongly anisotropic optical properties) and found that at $T > 1600$ K the difference is small ($\sim 10\%$) (Fig 2). We added an appropriate disclosure to the main text (Lines 196-200) and the supplementary material (Lines 188-201) of the resubmitted manuscript.

4. *Line 106: We wonder how difficult it would be to translate all the work done in this study from 2D to 3D (i.e., model thermal diffusivity in 3D and apply to rectangular slab model), and whether this would affect the final conclusions.*

We thank Reviewer #2 for posing this question.

1) The choice of dimensionality for the model is based on the tradeoff between the resolution and computational time. In 2D we were able to resolve very well the heat diffusion equation (500 m grid space) while maintaining a sustainable computational time (~10 h per model), allowing to run the simulations on a PC. This computational efficiency allows to quickly explore the parameter space (e.g. slab age, sinking velocity, A_{lat} -only and A_{rad} -including cases) by running several thermal evolution models in a short time span. Creating a 3D model by adding slab width while maintaining the same size and resolution of its thickness, would increase the computational time by a factor of ~240 (on the same PC). In the future, we will use advanced computer clusters to test the implications of our results in 3D.

2) We expect, however, that the results of our 2D models are robust and will not change for 3D models. As the number of dimensions increases, it increases also the directions from which the heat flow is established. An initially cold 3D body surrounded by a hot environment is unavoidably heated up faster than in the 2D case. Consequently, the faster heating in the 3D model will enhance the difference between A_{lat} -only and A_{rad} -including models. However, in the case examined in this study (i.e., subducting slab) we expect that the relative difference between the two sets of models will not change significantly. In our simulations, we solved the heat diffusion along the slab thickness, which is its shortest dimension and the one with the strongest temperature gradient AT (i.e., the most relevant dimension for diffusion problems). The width of a slab segment (3rd dimension) is typically much longer than its thickness (up to 10^3 km) and the temperature gradient along this dimension is significantly weaker. The main conclusion of this work, therefore, is robust. We thank Reviewer #2 for these questions, and we add an appropriate statement to the supplementary material of the resubmitted manuscript (Lines 442-457).

5. *Lines 110-113: These assumptions about the modal abundance of olivine in the slab are well-justified, however, is it reasonable to then assume predictions about the slab temperature increase represents a sort of maximum estimate? For example, in a slab with only 60% olivine (instead of 100%) the increase in temperature due to consideration of rad (reported as 100-200K by the authors) may be significantly less if the rad contribution by the other 40% of the slab is relatively lower.*

We thank Reviewer #2 for asking these interesting questions. Indeed, our measurements give an upper boundary estimate of the aggregate thermal conductivity of upper mantle rocks. The presence of opaque minerals may significantly reduce the radiative thermal conductivity of the whole rock. However, the radiative thermal conductivities of orthopyroxenes (opx), clinopyroxenes (cpx), and garnets (grt) are expected to be similar to olivine's at $T > 1500$ K (Grose & Afonso 2019). Therefore, the aggregate radiative conductivity of pyrolite (ol, opx, cpx, grt) and metabasaltic crust (opx, cpx, grt) computed using Grose & Afonso 2019 estimates at the P, T conditions of the upper mantle are in the range of $1 < A_{rad} < 3.5$ $W_{m^{-1}} K^{-1}$ (i.e., compatible to that of olivine). We thank Reviewer #2 for these questions, and we added an appropriate statement to the main manuscript (Lines 159-163) and to the supplementary material (Lines 485-506).

6. *Line 148: Although accessible from the data, it would be very useful to the reader to provide an in-text expression for $A_{rad} OI \sim X A OI$ values such that the reader can determine X for a series of different depths, e.g., $X = 0.4$ for a depth of 410km.*

We added a new figure in the Supplementary Materials, Fig. S6, that shows the ratios A_{rad}/A_{lat} and A_{rad}/A_{tot} computed along the mantle adiabat. To calculate the ratios for any P, T condition one can simply use equations S6-S9 with the coefficients provided in Extended Data Table 2 of the Supplementary Materials.

7. *Lines 203-205: This sentence about how the radiative heat transport may promote slab bending and stagnation can probably be removed as it is repeated again on lines 214-215.*

We agree with Reviewer #2, and we have removed the lines 203-205 to avoid redundances.

8. *For the reference n. 40 (MATLAB) it could be useful to write the version of MATLAB used in the main text.*

We agree with Reviewer #2, and we reported the version number also in the main text (see Line 138).

9. *Not clear why for reference n. 41 in the reference list, the authors added the DOI number, which is missing for all other references.*

We thank Reviewer #2 for pointing out the inconsistency of reporting just one DOI number. We decided to include the DOI in all references, whenever it is present, to ensure the discoverability of the reference article and enhance the citation accuracy.

10. *Please have a look at the format for all references in the reference list, sometimes we see the issue number, sometimes this number is not reported.*

We thank Reviewer #2 for reporting on the absence of the issue number. We thoroughly checked each reference, and we added the missing issue number in: Ref.8 (line 349) and Ref.9 (line 351).

Reviewer #3 (Remarks to the Author):

We thank Reviewer #3 for reviewing our manuscript.

As is documented below, we have incorporated the constructive suggestions of Reviewer #1 in our manuscript, and we have answer to the Reviewer's concerns and questions. Below, all reviewers' comments are in italics and our point-by-point responses are in normal fonts with blue colour. Changes in our revised main manuscript are also labelled by blue colour.

Reviewer #1 (Remarks to the Author):

The authors did a good job in addressing my concerns, especially in extending the Supplementary material with more detailed descriptions of the model setup and its limitations. I have only a few additional comments.

We thank Reviewer #1 for accepting our revised manuscript and providing further constructive comments.

1. *I think that the error estimate of the presented experimentally-derived radiative conductivity still needs a bit more explanation. As a reader, I would like to have some insight into why 30% is the right value (and not, e.g., 3% or 50%). Were there other studies with a similar technique that came to this value? Or does it come from your experience with the experimental setup? I understand that it is difficult to rigorously calculate the error, but I suppose there is some standard way to deal with that.*

We thank Reviewer #1 for this suggestion. We edited the main text accordingly to provide more details on this error estimate (Lines 152-156):

“The relative error of our estimates is $\Lambda_{rad}^{0l} \pm 30\%$. This is an empirical estimate that takes into account how the uncertainty in thickness and the temperature of the sample under high P [37-39] propagate to the error in the absorption coefficient. The uncertainty in temperature is particularly important because the absorbance of the sample is measured across a volume subject to a strong temperature gradient along the laser-heating direction.

2. *To include your T -dependent radiative conductivity in a subduction model, I would ideally have to follow the procedure described in the Supplement (S6). It is possible, but not straightforward, compared to, for example, analytical formulas provided by other authors. Could you suggest a more straightforward way to use your data? It seems to me that a linear interpolation between the data points in the Extended Data Table 1 will work fine in many applications. An explicit reference to this table may be useful in the main text. It would give a reader a hint on how to use your results without reading the entire supplement. It may increase the impact of your work!*

We thank Reviewer #1 for raising this point. We agree that the formulation of algorithm (S6) is not immediately intuitive. However, we designed it deliberately to ensure numerical stability and physical consistency in solving the heat diffusion equation (S27). We would like to emphasize that the raw data used to derive algorithm (S6) were already provided in previous versions of the manuscript (Fig. S5 and Extended Data Table S1). This was intended to offer transparency and flexibility, allowing other researchers to derive their own Λ_{rad} formulation.

In response to the Reviewer #1's suggestion we have added an explicit reference to Extended Data Table S1 in the main text (Lines 292–293), to guide readers toward the Λ_{rad} dataset. Moreover, to broaden the accessibility of our work, we have now included, in the SM, a more straightforward formulation of Λ_{rad} alongside the original algorithm (Lines 218-225; caption of Fig. S5; Extended Data Table S1).

*“Alternatively, $\Lambda_{rad}(T)$ can be computed with a more straightforward formulation by employing a 6th-order polynomial $\Lambda_{rad} = \sum_{i=0}^6 b_i T^i$, extrapolated from the dataset reported in Fig. S5 and Extended Data Tab.1. The algorithm (S6) **Error! Reference source not found.** employed in our numerical models, however, is bounded between minimum (Λ_{rad}^{bot}) and maximum values (Λ_{rad}^{top}), and is thus designed to self-consistently avoid, for any value of T , negative or unrealistic values of Λ_{rad} which may compromise the solution of the heat diffusion equation (S27). We therefore recommend using the 6th-order polynomial to compute Λ_{rad} , only for the limited temperature range of $298 < T < 2000$ K.”*

3. *I attach a pdf of the manuscript with a few minor corrections and suggestions that you may consider.*

We thank Reviewer #1 for the suggestions; we have gladly incorporated them in our manuscript.

PDF comments:

4. *Line 75: Here and elsewhere: "P, T" and "P-T" is not used consistently*

We have substituted "P-T" with "P, T" everywhere in the manuscript.

5. *L195-196: This sentence needs polishing. Perhaps: "... arise from the limited spectral coverage and/or temperature range of the latter"?*

The new sentence now reads:

*“The major differences between our α^{0l} measurements and most previous room-P, high-T studies^[28-30] arise from their limited spectral coverage^[30], the **investigated** T range (e.g, $T < 673$ K^[28]), or both^[29].”*

6. *L243: I am not sure why Fig. 2 is referenced here.*

We moved the reference to Fig.2, which now refers to the sentence before the comma:

“Therefore, as temperature increases, radiative heat transport is enhanced and overcompensates for the reduction of lattice conductivity induced by T (Fig. 2), ...”

7. *Fig3: Visually, panels a) and b) are practically the same. Their difference is captured in panel c). I would prefer to know the conductivity in the slab - Panel a) may be replaced with*

the conductivity plot (like Fig. S17a) that you also need in the description of the model (page 12).

We acknowledge Reviewer #1's suggestion. However, we respectfully disagree with implementing the suggested change. Fig. 3 shows the thermal fields of the two cases in panels (a) and (b), i.e., with and without radiative heat transport, the primary results of our work. Panel (c) highlights the differences between panels (b) and (a). The suggested change to display different quantities in panels (a) and (b) would make it difficult to the reader to understand what is shown in panel (c).

Furthermore, presenting thermal conductivity distributions in panel (a), as suggested by Reviewer #1, can be misleading and must not be used to read off the conductivity of the slab because conductivity profiles are highly sensitive to the temperature structure of the slab, which is itself model-dependent. For instance, changes in slab velocity or age would yield significantly different conductivity distributions. The variation of thermal conductivity with depth (i.e., along the P , T profile of the mantle), is already provided in Figure 2, where is presented more clearly.

8. *Fig. 4: Add a reference to an appropriate section or figure in the Supplement?*

We added a reference to Fig. S13 in Fig.4 (i.e., P-T path of the slab models crossing phase reactions).

9. *L312: We therefore propose/suggest/argue that...*

The new sentence now reads:

“We therefore suggest that future numerical models of slab subduction should account for radiative heat transport.”

Supplement:

10. *Fig. S6 (and elsewhere): is the word "adiabatic" appropriate. I assume that in the shallow part you prescribe a lithospheric geotherm. Caption of Fig. S13: Fig. 3.b -> Fig. 3?*

The upper 40 km of the P , T profiles in Fig. S6 and elsewhere have been computed using a linear temperature gradient of 35 K/km , which is typical of the lithosphere. Therefore, the word “adiabatic” was not sufficient to describe these P , T profiles. We have explicitly mentioned the 35 K/km gradient in the SM at:

- Lines 251-252
- Fig. S6
- Line 258
- Line 359
- Fig S7

In Fig S13, however, the word adiabatic is appropriate because the orange line represents the adiabatic P , T profile reported by Ref. [54]. Moreover, as stated in lines 441-443 of the SM, in our models we did not use the linear temperature gradient to compute the temperature of the

environment outside the slab. Therefore, there is no need to mention the linear temperature gradient of 35 K/km in Fig. S13 and Fig 3.

Reviewer #2 (Remarks to the Author):

I am happy with the revisions carried out by the authors. In my opinion, they addressed all concerns/questions/suggestions by the reviewers. I don't request further changes.

We thank Reviewers #2 and #3 for providing constructive comments in the previous review. We appreciate their expertise, which helped us improve our manuscript.